

# Comparing general and specialized word embeddings for biomedical named entity recognition

Rigo E. Ramos-Vargas, Israel Román-Godínez and
Sulema Torres-Ramos

Departamento de Ciencias Computacionales, Universidad de Guadalajara, Guadalajara,
Jalisco, México

## ABSTRACT

Increased interest in the use of word embeddings, such as word representation, for biomedical named entity recognition (BioNER) has highlighted the need for evaluations that aid in selecting the best word embedding to be used. One common criterion for selecting a word embedding is the type of source from which it is generated; that is, general (e.g., Wikipedia, Common Crawl), or specific (e.g., biomedical literature). Using specific word embeddings for the BioNER task has been strongly recommended, considering that they have provided better coverage and semantic relationships among medical entities. To the best of our knowledge, most studies have focused on improving BioNER task performance by, on the one hand, combining several features extracted from the text (for instance, linguistic, morphological, character embedding, and word embedding itself) and, on the other, testing several state-of-the-art named entity recognition algorithms. The latter, however, do not pay great attention to the influence of the word embeddings, and do not facilitate observing their real impact on the BioNER task. For this reason, the present study evaluates three well-known NER algorithms (CRF, BiLSTM, BiLSTM-CRF) with respect to two corpora (DrugBank and MedLine) using two classic word embeddings, GloVe Common Crawl (of the general type) and Pyysalo PM + PMC (specific), as unique features. Furthermore, three contextualized word embeddings (ELMo, Pooled Flair, and Transformer) are compared in their general and specific versions. The aim is to determine whether general embeddings can perform better than specialized ones on the BioNER task. To this end, four experiments were designed. In the first, we set out to identify the combination of classic word embedding, NER algorithm, and corpus that results in the best performance. The second evaluated the effect of the size of the corpus on performance. The third assessed the semantic cohesiveness of the classic word embeddings and their correlation with respect to several gold standards; while the fourth evaluates the performance of general and specific contextualized word embeddings on the BioNER task. Results show that the classic general word embedding GloVe Common Crawl performed better in the DrugBank corpus, despite having less word coverage and a lower internal semantic relationship than the classic specific word embedding, Pyysalo PM + PMC; while in the contextualized word embeddings the best results are presented in the specific ones. We conclude, therefore, when using classic word embeddings as features on the BioNER task, the general ones could be considered a good option. On the other hand, when using contextualized word embeddings, the specific ones are the best option.

Corresponding author
Sulema Torres-Ramos,
sulema.torres@cucei.udg.mx

## INTRODUCTION

Word embeddings which are representations of words as numerical vectors (*Mikolov et al., 2013*), have become one of the most useful features for several natural language processing (NLP) tasks (*Ghannay et al., 2016*), as they have been used for machine translation (*Mikolov et al., 2013*; *Qi et al., 2018*), question answering (*Bordes, Chopra & Weston, 2014*; *Dimitriadis & Tsoumakas, 2019*), and sentiment analysis (*Altowayan & Tao, 2016*; *Rezaeinia et al., 2019*), among others (*Mohd, Jan & Shah, 2020*). In the specific case of the named entity recognition (NER) task the use of word embeddings is recommended to avoid time-consuming feature engineering (*Unanue, Borzeshi & Piccardi, 2017*). The huge increase in the use of word embeddings is due to their ability to capture semantic properties and linguistic relationships between words. They can be trained using either specialized sources (e.g., articles, books, scientific databases, etc.) or general sources (e.g., Wikipedia, Twitter). Some examples of specialized word embeddings are Pyysalo PM + PMC (*Moen & Ananiadou, 2013*), Chiu win-30 (*Yepes & MacKinlay, 2016*), and Aueb dim-400 (*McDonald, Brokos & Androutsopoulos, 2018*), while some general word embeddings are Glove Wiki + GW (*Mikolov et al., 2013*), Glove CC-840 (*Mikolov et al., 2013*), and Glove Twitter (*Mikolov et al., 2013*).

To date, no consensus has been reached as to which kind of word embedding is best for each NLP task, so selection is based on comparing various experimental setups and each researcher's own experience. In the case of biomedical named entity recognition (BioNER), which intents to identify "chunks of text that refer to specific entities of interest such as gene, protein, drug and disease names" (*Campos, Matos & Oliveira, 2012*), several authors recommend specialized word embeddings (*Unanue, Borzeshi & Piccardi, 2017*; *Batbaatar & Ryu, 2019*), but this raises two questions: can general word embeddings be recommended for use with BioNER?; and, is it always necessary to train specialized word embeddings for each BioNER context?

As the analysis developed in this study shows, the state-of-the-art approach supports specific word embeddings for some NLP tasks, including BioNER, because this type offers greater coverage of domain-specific words and maintains better quality of the semantic and syntactic relationships. However, many works in this field have focused on two aspects: (a) studying the behavior of the word embeddings (general vs. specific) applied to the BioNER task using only one NER algorithm, and, (b) studying the performance of the algorithms on the BioNER task by combining the word embeddings with additional features. For these reasons, the present study was designed to evaluate the classic word embeddings GloVe Common Crawl and Pyysalo PM + PMC (general and specific, respectively), and the contextualized word embeddings ElMo, Pooled Flair, and Transformer (with their general and specific version) as the only feature extractor for addressing the BioNER task in a drug context, over two corpora, DrugBank and MedLine. In addition, for classic word embeddings three of the most common NER algorithms:

conditional random fields (CRF), bidirectional long short-term memory (BiLSTM), and a combination of the two (BiLSTM-CRF), were used; on the other hand, for contextualized word embeddings only the BiLSTM-CRF algorithm was used. Adopting this approach allowed us to test some state-of-the-art hypotheses for choosing between general and specific classic word embeddings, and to determine that the specific form is not always better than the general one for performing the BioNER task; also, between general and specific contextualized word embeddings, the latter seems to be the best option.

## Related work

*Wang et al. (2018)* elaborated intrinsic and extrinsic comparisons using four word embeddings from clinical notes, biomedical publications, Wikipedia, and Google news. Their intrinsic evaluation found that the word embeddings generated from the biomedical literature had better semantic relations than those taken from general texts. However, upon conducting an extrinsic evaluation that depended on the NLP task, they found that word embeddings trained from biomedical sources do not necessarily show better performance than those trained from general sources, since better results were obtained on the relation extraction (RE) task using the general word embedding from Google news. One drawback of their study, however, is that the word embeddings were not evaluated on the NER task.

*Segura-Bedmar, Suárez-Paniagua & Martnez (2015)* evaluated two word embeddings, one trained from Wikipedia general texts, the other from PubMed biomedical literature, in order to improve the performance of the CRF algorithm on the NER task. Surprisingly, the algorithm trained with the word embedding from Wikipedia obtained the best score. Though only one algorithm was used, additional features were added, such as part of speech (POS), context windows, and orthographic features.

A wider comparison of word embeddings was carried out by *Batbaatar & Ryu (2019)*, as they evaluated five general and nine specific word embeddings to observe their impact on the BioNER task. In this case, results showed that all the specific word embeddings obtained the best scores, so they recommended using this approach for BioNER. The general embeddings used in their work were Glove Wiki + GW, Glove CC-42, Glove CC-840, Glove Twitter, and Word2Vec, while the specific ones were Pyysalo PM, Pyysalo PMC, Pyysalo PM + PMC, Pyysalo Wiki PM + PMC, Chiu win-2, Chiu win-30, Chen PM + MIMIC III, Aueb dim-200, and Aueb dim-400. However, three aspects considered in their evaluation impeded observing the real impact of word embeddings on performance: (a) the use of other features, like character embedding and POS; (b) an automatic evaluation using the UMLS metathesaurus (Unified Medical Language System) instead of a manual gold standard; and (c) the use of the BiLSTM-CRF algorithm alone.

A variant of the aforementioned works was developed by *Unanue, Borzeshi & Piccardi (2017)*, who used different models along with several combinations of characteristics, including word embeddings, to perform drug-named recognition in the DrugBank and MedLine corpora. The models used were CRF, BiLSTM, and BiLSTM-CRF, while the word embeddings were Common Crawl and Common Crawl + MIMIC-III (CC/mimic). Three important aspects were found: (a) the best model was BiLSTM-CRF; (b) the best

word embedding was CC/mimic; and (c) the greater the coverage of words in the word embeddings, the better the performance. However, since CC/mimic cannot be considered completely specialized because it is the concatenation of Common Crawl (general embedding) and MIMIC-III (specific embedding), an additional study is necessary to observe the difference between general vs. specific word embeddings.

The above-mentioned works compared classic word embeddings, which are static pre-computed vectors, however, novel embeddings techniques, known as contextualized, generate dynamic vectors based on the context words (*Peters et al., 2018a*, *2018b*; *Yamada et al., 2020*), allowing to address the polysemous and context-dependent nature of the words. As a result, some studies have focused on improving the performance of the NER task with contextualized embeddings (*Akbik, Blythe & Vollgraf, 2018*; *Akbik, Bergmann & Vollgraf, 2019*) and more recently on the BioNER task (*Patel, 2020*). The latter, however, has combined both classical and contextualized embeddings with their respective general and specific version achieving the best results in multiple biomedical corpora. Also, *Jiang, Sanger & Liu (2019)* have studied the effects of combining multiple contextualized embeddings. But, to the best of our knowledge, there is no study comparing general vs. specific contextualized embeddings.

In summary, no comparison of general vs. specific classic word embeddings has yet been elaborated for BioNER that (i) avoids the use of additional features; (ii) uses several algorithms; and (iii) considers the coverage and semantic relationships of the word embeddings. Also, a comparison between general and specific contextualized word embeddings on the BioNER has not yet been presented.

# MATERIALS AND METHODS

This section describes the corpora, word embeddings, models, parameter settings, and metrics used in the experiments described above.

## Corpora

To train the models, two semantically annotated corpora were selected: DrugBank and MedLine. Both were used as the gold standard on the SemEval-2013 DDI Extraction Task; specifically, task 9.1 for the recognition and classification of pharmacological substances (*Herrero-Zazo et al., 2013*). These corpora have been widely used for BioNER (*Segura-Bedmar, Suárez-Paniagua & Martnez, 2015*; *Chalapathy, Borzeshi & Piccardi, 2016*; *Unanue, Borzeshi & Piccardi, 2017*) because they provide a common framework for comparing the latest advances in this task. The corpora have been manually annotated by experts in pharmacovigilance and each contains four entity types: drugs, groups, brands, and substances not approved for human use (NH). Figure 1 depicts an example of a DrugBank's document annotated according to the SemEval-2013 DDI Extraction Task, where the XML elements identify several parts of the document, such as sentences and entities of interest with theirs corresponding entity type. In this particular example, we can observe the sentence "Bromocriptine mesylate may interact with dopamine antagonists, butyrophenones, and certain other agents.", which contains three bioentities:

```
−<sentence id="DDI-DrugBank.d272.s1" text="Bromocriptine mesylate may
  interact with dopamine antagonists, butyrophenones, and certain other
  agents.">
    <entity id="DDI-DrugBank.d272.s1.e0" charOffset="0-21" type="drug"
    text="Bromocriptine mesylate"/>
    <entity id="DDI-DrugBank.d272.s1.e1" charOffset="41-60" type="group"
    text="dopamine antagonists"/>
    <entity id="DDI-DrugBank.d272.s1.e2" charOffset="63-76" type="group"
    text="butyrophenones"/>
    <pair id="DDI-DrugBank.d272.s1.p0" e1="DDI-DrugBank.d272.s1.e0"
    e2="DDI-DrugBank.d272.s1.e1" ddi="true" type="int"/>
    <pair id="DDI-DrugBank.d272.s1.p1" e1="DDI-DrugBank.d272.s1.e0"
    e2="DDI-DrugBank.d272.s1.e2" ddi="true" type="int"/>
    <pair id="DDI-DrugBank.d272.s1.p2" e1="DDI-DrugBank.d272.s1.e1"
    e2="DDI-DrugBank.d272.s1.e2" ddi="false"/>
</sentence>
```

**Figure 1 Example of an annotated sentence of the DrugBank corpus according to SemEval-2013 DDI Extraction Task.**

**Table 1 Statistics of the training and test corpora used in the experiments.** Obtained from *Unanue, Borzeshi & Piccardi (2017)*.

| | DrugBank | | MedLine | |
|---|---|---|---|---|
| | Training | Test | Training | Test |
| Documents | 730 | 54 | 175 | 58 |
| Sentences | 6,577 | 145 | 1,627 | 520 |
| Drug | 9,715 | 180 | 1,574 | 171 |
| Group | 3,832 | 65 | 234 | 90 |
| Brand | 1,770 | 53 | 36 | 6 |
| NH | 124 | 6 | 520 | 115 |

*bromocriptine mesylate*, *dopamine antagonists*, and *butyrophenones*, labeled as entities with their corresponding type: *drug*, *group*, and *group*, respectively.

In addition, the corpora are divided into two datasets: training and test. Those datasets are disjoint (i.e., the documents for training are different from those for testing) and contain files with the same XML structure. Table 1 describes the statistics related to these corpora. The first two rows show the number of documents and sentences included in each corpus for both the training and testing stages. DrugBank, for example, has 730 documents for training and 54 for testing, while MedLine has 175 and 58 for training and testing, respectively. The last four rows show the number of biomedical entities contained in each corpus. Here, DrugBank has 9,715 drug entities for training and 180 for testing, while MedLine has 1,574 and 171, respectively, for training and testing.

## Word embeddings

Word embedding maps words into low-dimensional numerical vectors by capturing syntactic and semantic similarities (*Mikolov et al., 2013*). The methods used to generate embeddings include probabilistic models (*Globerson et al., 2007*), neural networks

(*Mikolov et al., 2013*), dimensional reduction on the word co-occurrence matrix (*Levy & Goldberg, 2014*), and explainable knowledge (*Qureshi & Greene, 2019*).

The first word embeddings selected for the experiments were GloVe Common Crawl (CC) and Pyysalo PMC + PM (Pyy). Both embeddings are considered *classic* because they are static and word-level, resulting on one pre-computed embedding per word (*Akbik, Blythe & Vollgraf, 2018*). CC has often been used for general domains (*Lester et al., 2020*; *Ronran & Lee, 2020*), while Pyy has shown the best scores for pharmacological substances (*Batbaatar & Ryu, 2019*). GloVe Common Crawl is a general word embedding released in 2014 by the computer science department at Stanford University (*Pennington, Socher & Manning, 2014*), trained by the Global Vectors method (GloVe) through 840 billion general tokens on the web that generated a vocabulary of close to 2.2 million unique concepts with 300 dimensional vectors. Pyysalo PMC + PM, in contrast, is a specific word embedding released in 2013 to aid in linguistic tasks related to the field of biomedicine (*Moen & Ananiadou, 2013*). It was trained with the word2vec method (*Mikolov et al., 2013*) through 5.5 billion specific tokens from 14 million PubMed abstracts and 700,000 PubMed Central articles, generating a vocabulary of close to 4 billion unique concepts with 200 dimensional vectors.

Classic word embeddings can be compared in different ways (*Wang et al., 2018*) since they generate static vectors. However, it is important to analyze the most recent general and specific word embeddings, extracted with state-of-the-art techniques, which generate context-dependent dynamic vectors. For this reason, a general and specific version of ELMoEmbeddings, PooledFlairEmbeddings, and TransformerWordEmbeddings were used.

ELMo embeddings were presented by *Peters et al. (2018a)*. They are a contextualized word representation that models characteristics of word use (e.g., syntax and semantics), and how these uses vary across linguistic contexts (i.e., to model polysemy). Their vectors are functions learned from internal states of a deep bidirectional language model (biLM) pre-trained on a large text corpus. The general and specific embeddings selected are Original and PubMed, respectively. The Original embedding was trained on the 1 Billion Word Benchmark, while the PubMed embedding was trained with sentences from abstracts of the PubMed database. Both embeddings had the same parameters, as can be verified on their web page (https://allennlp.org/elmo).

Unlike ELMo embeddings, which are a word-level representation, Flair embeddings are a contextualized character-level representation. Flair embeddings, presented by *Akbik, Blythe & Vollgraf (2018)*, are obtained from the hidden states of a forward-backward recurrent neural network. They are trained without any explicit notion of words, instead, they model words as sequences of characters. Also, these embeddings are contextualized by their surrounding text, meaning that the same word will have different embeddings depending on its contextual use. The variants of the Flair embeddings used in this study are the Pooled Flair embeddings (*Akbik, Bergmann & Vollgraf, 2019*), which consume more memory resources but perform better. Pooled Flair embeddings evolve over time obtaining different embeddings of the same words in the same sentence at two different points in time. The general and specific embeddings selected from Pooled Flair are Mix

and PubMed, respectively. The Mix embedding is trained with general corpus from the web, Wikipedia, and subtitles. On the other hand, PubMed embedding is trained with 5% of the PubMed abstracts until 2015.

Finally, Transformer word embeddings, introduced by *Peters et al. (2018b)*, are generated through an adaptation of the Transformer architecture to bidirectional language modeling. The Transformer is a feed forward self-attention based architecture widely used for multiple NLP tasks (*Vaswani et al., 2017*). The transformer embeddings selected are BERT and BioBERT (general and specific, respetively). BERT was pre-trained using the BooksCorpus (800M words) and English Wikipedia (2,500M words) (*Devlin et al., 2018*). On the other hand, BioBERT was pre-trained on biomedical domain corpora using PubMed abstracts and PMC full-text articles (*Lee et al., 2020*).

## Models

The algorithms selected to tag the drug entities were conditional random fields (CRF), BiLSTM, and BiLSTM-CRF. The CRF algorithm was selected because it is a statistical algorithm commonly used for BioNER (*Settles, 2004*; *Rais et al., 2014*; *Segura-Bedmar, Suárez-Paniagua & Martnez, 2015*). CRF has demonstrated better performance on the NER task than other statistical techniques like the hidden Markov (HMM), support vector machine (SVM), and maximum entropy Markov models (MEMM) (*Campos, Matos & Oliveira, 2012*). In addition, two recurrent neural networks (RNN) were selected: BiLSTM and BiLSTM-CRF, both of which are state-of-the-art and have achieved some of the best scores for BioNER (*Unanue, Borzeshi & Piccardi, 2017*; *Batbaatar & Ryu, 2019*).

### Conditional random field

Conditional random fields is a probabilistic discriminative model presented by *Lafferty, McCallum & Pereira (2001)* that belongs to the set of undirected graphic models and works with conditional probability, $p(\vec{y}|\vec{x})$, to predict tag sequences, $\vec{y}$, through observations, $\vec{x}$, taking into account the data sequence (*Lafferty, McCallum & Pereira, 2001*). The equation for the CRF model is:

$$P(\vec{y}|\vec{x}) = \frac{1}{Z(\vec{x})} \exp\left( \sum_{j=1}^{n} \sum_{i=1}^{m} \lambda_i f_i(y_{j-1}, y_j, \vec{x}, j) \right) \tag{1}$$

where $f$ is the set of feature functions with the values of the input vector, $\vec{x}$, the data predicted position, $j$, and the tags in positions $j-1$ and $j$. Each set of feature functions has a weight, $\lambda$, and the equation is normalized through $Z(\vec{x})$ factor, as follows:

$$Z(\vec{x}) = \sum_{y \in \vec{y}} \exp\left( \sum_{j=1}^{n} \sum_{i=1}^{m} \lambda_i f_i(y_{j-1}, y_j, \vec{x}, j) \right) \tag{2}$$

The model is trained by estimating the parameters $\lambda's$ with the maximum likelihood.

$$\hat{y} = *\mathrm{argmax}_{\vec{y}} P(\vec{y}|\vec{x}) \tag{3}$$

Finally, the Viterbi algorithm *Viterbi (1967)* is used to decode the optimal output sequence.

### Bidirectional long short-term memory

The long short-term memory presented by *Hochreiter & Schmidhuber (1997)* is a kind of artificial RNN used to classify and process sequential data. This RNN solves the long-term memory problem caused by gradient vanishing using cells with different gates to regulate the information flow. Eqs. (4)–(7) of the cell are presented below:

$$i_t = \sigma(W_{xi}x_t + W_{hi}h_{t-1} + W_{ci}c_{t-1} + b_i) \tag{4}$$

$$c_t = (1 - i_t)c_{t-1} + i_t\tanh(W_{xc}x_t + W_{hc}h_{t-1} + b_c) \tag{5}$$

$$o_t = \sigma(W_{xo}x_t + W_{ho}h_{t-1} + W_{co}c_t + b_o) \tag{6}$$

$$h_t = o_t\tanh(c_t) \tag{7}$$

This version does not have a forget gate ($f$), but only input ($i$) and output ($o$) gates, as in the work of *Lample et al. (2016)*. The cell vectors are represented by $c$, the hidden vectors by $h$, and the weights and bias by $W$ and $b$, respectively. The bidirectional version of this model was selected because it can access both past and future input data (*Graves, Mohamed & Hinton, 2013*). In this variation, the hidden vectors are generated in two ways: left to right $\overrightarrow{h}$, and right to left $\overleftarrow{h}$, then the hidden vectors are concatenated to obtain the final representation: $h_t = [\overrightarrow{h} + \overleftarrow{h}]$ (*Unanue, Borzeshi & Piccardi, 2017*).

### Bidirectional LSTM-CRF

*Lample et al. (2016)* presented an improved version of the LSTM network using a CRF output layer instead of a softmax output layer. This layer, described previously in the CRF model section, was used to take into account neighboring tags. The hidden vectors $h = (h_1, h_2, ..., h_n)$ were used as the input for the CRF layer to predict the output label sequence $y = (y_1, y_2, ..., y_n)$.

## Experimental setup

In order to compare general and specific word embeddings on the BioNER task, four experiments were proposed; the first three experiments correspond to classic word embeddings while the last experiment corresponds to contextualized ones. The first experiment was designed to evaluate the performance obtained between general and specific classic word embeddings using three models; the second, to test the impact on performance for classic word embeddings when different sizes of training corpus were used; the third, to observe the influence that intrinsic semantic relationships of the classic word embeddings have on NER performance; and the fourth, to evaluate the performance obtained between general and specific contextualized word embeddings.

### First experiment: general vs. specific classic word embeddings

This experiment consisted in testing the CRF, BiLSTM, and BiLSTM-CRF models using classic word embeddings as the only feature. The objective was to identify the classic word embedding that yielded the best results (general vs. specific) on the BioNER task. First, each corpus was split into sentences and tokens, then each token was labeled according to the IOB scheme (*Ramshaw & Marcus, 1999*) to represent whether or not the word

**Table 2 Sentence tagged using IOB format.**

| Tokens | IOB Format class |
|---|---|
| Bromocriptine | B-Drug |
| mesylate | I-Drug |
| may | O |
| interact | O |
| with | O |
| dopamine | B-Group |
| antagonists | I-Group |
| butyrophenones | B-Group |
| and | O |
| certain | O |
| other | O |
| agents | O |
| . | O |

belonged to an entity class. Here, the letter B (Beginning) indicates that a word is the initial part of a named entity, I (Inside) signals that the word is part of the named entity, and O (Outside) indicates that the word is not a part of a named entity. Table 2 shows a sentence tagged using IOB format, the first column indicates the sentence's tokens while the second column indicates the class of such tokens. In this example, the tokens *Bromocriptine, mesylate, dopamine, antagonists*, and *butyrophenones* correspond to biomedical entities. The MedLine and DrugBank corpora formatted with the IOB scheme are available in Data S1.

After formatting the corpora, one vector for each token was retrieved from the word embedding. These numerical vectors were used directly as features in the BiLSTM and BiLSTM-CRF models, but first they had to be transformed as dictionaries for the CRF model. In cases where no token existed in the word embedding, its numerical vector was generated randomly in the range [−1, 1].

Once the embeddings were obtained from CC and Pyy, the models were set up. The CRF model was implemented using the CRFsuite from scikit-learn (*Pedregosa et al., 2011*), while the healthNER project (*Jauregi, 2017*) was used for the BiLSTM and BiLSTM-CRF models. To train and validate the parameters of these neural networks, each training corpus was divided into two parts: the first, to train the internal parameters of the network (70%); the second, to validate the hyper-parameters (30%) (*Bergstra & Bengio, 2012*). The hyper-parameters, or values assigned manually to the neural networks, included the number of hidden nodes (100), the dropout rate (0.5), and the learning rate (0.1). The general parameters for all the models were the dimensions of the word embeddings (300 for CC, 200 for Pyy) and the number of epochs (100). These settings are shown in Table 3. Finally, the best model saved during the epochs was tested with unseen data from the test section of the corpus. The source code testing the CRF, BiLSTM, and BiLSTM-CRF models is available in Data S2.

**Table 3 Parameters for BiLSTM and BiLSTM-CRF models.**

| Parameters | Value |
|---|---|
| Word embedding dimension | 300 (CC)/200 (Pyy) |
| LSTM hidden layer dimension | 100 |
| Epochs | 100 |
| Dropout | 0.5 |
| Optimization | Stochastic gradient descendent |
| Learning rate | 0.1 |

### Second experiment: training set reductions

As Table 1 shows, in the first experiment the number of entities for training in DrugBank was approximately ten times higher than those used by MedLine, while the number of entities for testing was similar in both corpora. This situation was observed by *Unanue, Borzeshi & Piccardi (2017)* as well, who hypothesized that it could be one of the reasons MedLine generally achieved lower performance compared to DrugBank.

To evaluate the impact of the size of the training corpus on the BioNER task, multiple reductions were applied to the DrugBank corpus. Five training sets of different sizes were generated by removing sentences from the training and validation sections of the DrugBank corpus. Tokens were removed from the end to the beginning until 75%, 50%, 25%, 12.5%, and 6.25% of the total number of entities for training and validation were reached. The part of the corpus that corresponded to the test remained the same size. The source code for corpus reduction and each reduced dataset are available in Data S3.

Subsequently, for each training set, the same processing as in experiment one was followed using only the model with the best performance in the state-of-the-art (*Unanue, Borzeshi & Piccardi, 2017*; *Batbaatar & Ryu, 2019*); that is, BiLSTM-CRF.

### Third experiment: classic word embeddings' semantic evaluation

As mentioned earlier, specific word embeddings maintain a better intrinsic semantic relationship than general ones (*Wang et al., 2018*). Therefore, this experiment was designed to observe the semantic relationships in the CC and Pyy word embeddings. For this experiment, *Wang et al. (2018)* methodology was used. First, four datasets were selected: *Pedersen et al. (2007)*, *Hliaoutakis (2005)*, MayoSRS (*Pakhomov et al., 2011*), and UMNSRS (*Pakhomov et al., 2010*). Each data set contains pairs of medical words, and each pair has an associated number that indicates the semantic similarity assigned by a human expert. Later, the CC and Pyy vectors were extracted for each medical word in the datasets. In cases where a word was not found in the word embedding, the vector was generated randomly between [−1, 1]. In the next step, the cosine similarity between each pair of words $(w_1, w_2)$ was calculated using Eq. (8), where $\theta_1$ and $\theta_2$ are vector representations for the words:

$$\text{similarity } (w_1, w_2) = \frac{\theta_1 \cdot \theta_2}{||\theta_1|| \, ||\theta_2||} \tag{8}$$

Finally, Pearson's correlation was calculated for the similarity between the datasets obtained with the word embeddings and those indicated by the human experts. The gold standard datasets used for measuring the intrinsic semantic relationship and the source code to perform the comparison with the word embeddings are available in Data S4.

### Fourth experiment: general vs. specific contextualized word embeddings

In order to compare the performance of the general vs. specific contextualized word embeddings on the BioNER task, the Flair framework (*Akbik et al., 2019*) was selected. Flair implements the BiLSTM-CRF sequence labeling architecture proposed by *Huang, Xu & Yu (2015)*, which is similar to the one used in the first and second experiment. However, this architecture does contain the forget gate, as Eq. (9) shows:

$$f_t = \sigma(W_{xf}x_t + W_{hf}h_{t-1} + W_{cf}c_t + b_f) \tag{9}$$

Also, Flair has a text embedding library with simple interfaces that allows to use and combine different word embeddings, including ELMo embeddings, Flair embeddings, Transformer embeddings, among others. Its implementation is very simple, allowing to replicate the models proposed for this experiment. First, three types of contextualized word embeddings have been selected: ELMo embeddings (*Peters et al., 2018a*), Pooled Flair embeddings (*Akbik, Bergmann & Vollgraf, 2019*), and Transformer embeddings (*Peters et al., 2018b*). Then, for each type of word embedding, general and specific versions were used: for ELMo, the Original and PubMed embeddings; for Pooled Flair, Mix and PubMed embeddings; and for Transformer, the BERT and BioBERT embeddings. Later, the models were set up, the number of hidden nodes (100), the dropout rate (0.5), the learning rate (0.1), and the mini-batch size $\in \{1, 2, 4, 8, 16, 32\}$. The models were trained using vanilla SGD with no momentum for 100 epochs, choosing the model with the best F1-score among the different epochs. As in the first experiment, the corpora for training and test were DrugBank and MedLine, with the same partitioning. The source code to test the contextualized word embeddings is available in Data S5.

## Evaluation metrics

The performance of the models in experiments one, two and four is reported in terms of F1-scores, an important measure that represents the harmonic mean of precision and recall. "Precision is the percentage of named entities found by the learning system that are correct. Recall is the percentage of named entities present in the corpus that are found by the system. A named entity is correct only if it is an exact match of the corresponding entity in the data file" (*Sang & De Meulder, 2003*).

The "strict" evaluation method proposed by (*Segura Bedmar, Martnez & Herrero Zazo, 2013*) was used, in which both the entity class and its exact boundaries must match the expected values of the data set (*Nadeau & Sekine, 2007*; *Unanue, Borzeshi & Piccardi, 2017*). Specifically, precision, recall, and F1-score have been using in different named entity recognition tasks as in MUC-7 (*Chinchor & Robinson, 1997*) and CoNLL 2003 (*Sang & De Meulder, 2003*), since the main objective is to correctly identify a word that represents a

**Table 4 Drug Name Recognition using the CRF, BiLSTM, and BiLSTM-CRF models, along with the GloVe Common Crawl (CC) and Pyysalo PM + PMC (Pyy) word embeddings for the DrugBank and MedLine corpora.**

| Corpus | Model | Precision | Recall | F1-score |
|---|---|---|---|---|
| DrugBank | CRF + CC | 82.07 | 66.3 | 73.34 |
| | BiLSTM + CC | 85.62 | 87.3 | 86.45 |
| | BiLSTM-CRF + CC | 86.79 | 89.9 | 88.32 |
| | CRF + Pyy | 79.79 | 64.9 | 71.58 |
| | BiLSTM + Pyy | 83.01 | 84.36 | 83.68 |
| | BiLSTM-CRF + Pyy | 84.24 | 85.34 | 84.79 |
| Medline | CRF + CC | 62.3 | 33.04 | 42.18 |
| | BiLSTM + CC | 60.44 | 61.52 | 60.98 |
| | BiLSTM-CRF + CC | 71.54 | 60.18 | 65.37 |
| | CRF + Pyy | 69.29 | 31.21 | 43.03 |
| | BiLSTM + Pyy | 63.36 | 64.21 | 63.78 |
| | BiLSTM-CRF + Pyy | 75.75 | 67.79 | 71.55 |

bioentity while the correctness tagging of the surrounding words, as long as they are not bioentities, are out of interest.

In the third experiment, Pearson's correlation coefficient was used to observe the linear correlation between similarity scores from human judgments and those from word embeddings. The equation for this coefficient is shown in Eq. (10), where $n$ is the sample size, $x_i$ and $y_i$ are the individual sample points, and $\bar{x}$ and $\bar{y}$ are the sample means.

$$r = \frac{\sum_{i=1}^{n}(x_i - \bar{x})(y_i - \bar{y})}{\sqrt{\sum_{i=1}^{n}(x_i - \bar{x})^2 \sum_{i=1}^{n}(y_i - \bar{y})^2}} \tag{10}$$

## RESULTS

Regarding to the evaluation of classic word embeddings, Table 4 shows the findings that correspond to the comparison of the three different models "CRF, BiLSTM, and BiLSTM-CRF" using CC (general) and Pyy (specialized) as the unique feature, tested on the DrugBank and Medline corpora (first experiment). To evaluate the models' performance, precision, recall, and F1-score metrics were computed.

The most efficient model for drug-named entity recognition is BiLSTM-CRF, at 88.32% and 71.55% for DrugBank and MedLine, respectively. In contrast, the least efficient model is CRF, at 71.58% for DrugBank and 42.18% for MedLine. This is consistent with *Unanue, Borzeshi & Piccardi (2017)*, and shows that the CRF model requires more extensive feature engineering or other kinds of representations for the word embedding feature (*Segura-Bedmar, Suárez-Paniagua & Martnez, 2015*); for example, clustering. It is important to note that the BiLSTM models maintained a balance between precision and recall regardless of the form of word embedding (CC or Pyy) or the corpus (DrugBank or Medline). On the one hand, observations of the DrugBank corpus show that the

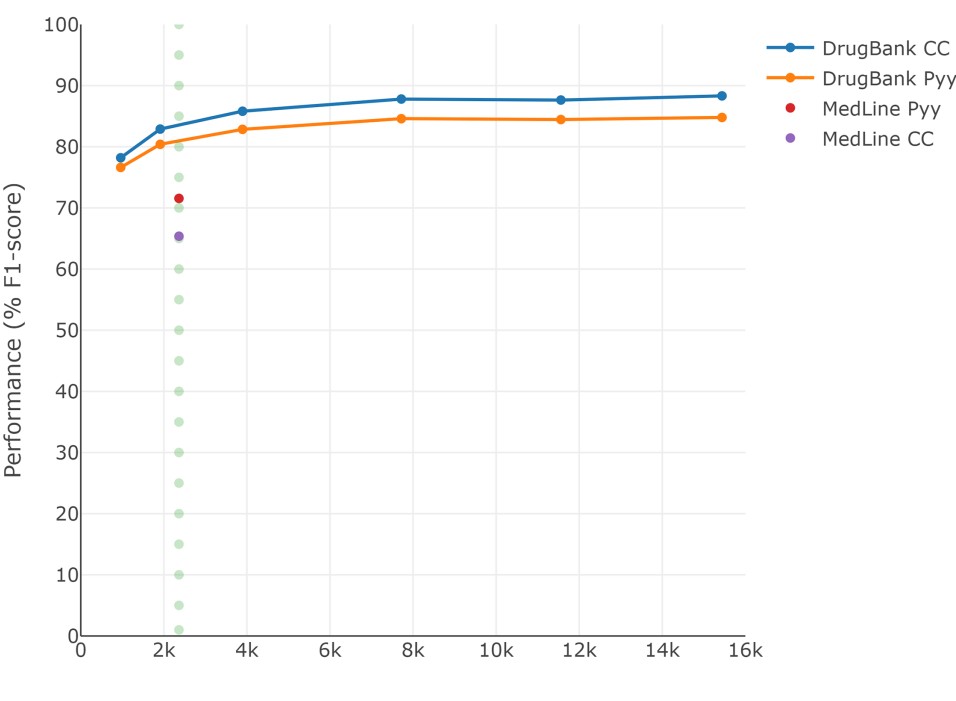

**Figure 2 Loss of DrugBank performance by reducing the size of the training corpus.**

differences between precision and recall using CC and Pyy are 1.68 and 1.35 percentage points (pp), respectively, while for the Medline corpus the differences between CC and Pyy are 1.08 pp and 0.85 pp. When the BiLSTM model was combined with CRF in the case of the DrugBank corpus, the difference between precision and recall is a little wider for CC at 3.11 pp, while for Pyy it is 1.1 pp. For Medline, the differences are considerably wider: CC at 11.36 pp and Pyy at 7.96 pp. However, these differences are related to the CRF layer appended to the BiLSTM, not to the corpora, because the wider differences between precision and recall were found for MedLine, while DrugBank maintained a balance. Based on the performance differences between the word embeddings in Table 4, and in order to determine whether these performances are related to the coverage of the word embeddings, Table 5 shows the percentage of coverage and F1-score obtained for the best model (BiLSTM-CRF) over both corpora, DrugBank and Medline, for each classic word embedding. With respect to DrugBank, the model with the specific word embedding Pyy had the best coverage (Pyy: 96.37% vs. CC: 93.93%), but not the best performance (Pyy: 84.79% vs. CC: 88.32%). For MedLine, the model with the specific word embedding Pyy had both the best coverage and the best performance.

Since the DrugBank training set is larger than the MedLine training set (see Table 1), and considering that training set size could be an important factor in performance (*Unanue, Borzeshi & Piccardi, 2017*), Fig. 2 presents the F1-scores obtained for each reduction over the DrugBank training set (second experiment). A performance loss of approximately 7 pp is visible when the DrugBank training set is reduced to similar

**Table 5 Comparison between coverage and performance with the BiLSTM-CRF model using the GloVe Common Crawl and Pyysalo PM + PMC word embeddings.**

| Word embedding | DrugBank | | MedLine | |
|---|---|---|---|---|
| | Coverage | F1-score | Coverage | F1-score |
| GloVe Common Crawl (CC) | 93.93 | 88.32 | 96.97 | 65.37 |
| Pyysalo + PM + PMC (Pyy) | 96.37 | 84.79 | 99.22 | 71.55 |

**Table 6 Pearson correlation coefficient between similarity scores from human judgments and those from classic word embeddings on four measurement datasets.**

| Dataset | GloVe Common Crawl (CC) | Pyysalo PM + PMC (Pyy) |
|---|---|---|
| Pedersen's | 0.446 | 0.575 |
| Hliaoutakis's | 0.292 | 0.283 |
| MayoSRS | 0.290 | 0.413 |
| UMNSRS | 0.363 | 0.420 |

conditions to those of MedLine, regardless of which of the two classic word embeddings is selected (dotted line), though the general word embedding CC continued to perform better than Pyy for DrugBank, despite the size reduction.

However, we still lacked an intrinsic evaluation of classic word embeddings on the BioNER task in order to observe the degree to which the automatically-created embeddings correlate with the gold standard. For this reason, we designed the third experiment, which allowed us to evaluate the affinity of each word embedding with respect to each biomedical corpus. Table 6 presents the Pearson correlation coefficient for the semantic similarities using the classic word embeddings and the judgments of human experts. Overall, the specific word embedding (Pyy) presents a better correlation with the human judgment for all datasets.

Finally, determining if the performance of general and specialized embeddings on the BioNER task is influenced by the technique used to generate them, Table 7 shows the mini-batch size, precision, recall, F1-score of the best performances obtained from testing the general and specialized version of each contextualized word embedding (ELMO, Pooled Flair, and Transformer) on the DrugBank and MedLine corpus. It is worth notice that for all cases the specialized version overcomes the general one. Also, using the DrugBank corpus, the best results are those presented with a larger mini-batch size (16 or 32), on the contrary, when using MedLine the mini-batch size are smaller (8 and 16). The complete results of the fourth experiment are available in Data S6.

## DISCUSSION

In the following paragraphs, we discuss first, the classic word embeddings, and then the contextualized ones. For the classic word embeddings, the results are presented for each corpus, first DrugBank, then MedLine; later, these two corpora are contrasted to observe and analyze differences in their performance.

**Table 7 Drug Name Recognition using the BiLSTM-CRF model, along with general and specific contextualized word embeddings for the DrugBank and MedLine corpora.**

| Corpus | Word embedding | mini-batch size | Precision | Recall | F1-score |
|---|---|---|---|---|---|
| DrugBank | ELMo Original | 32 | 82.29 | 86.31 | 84.26 |
| | ELMo PubMed | 16 | 87.03 | 89.58 | 88.28 |
| | Pooled Flair Mix | 32 | 81.9 | 86.97 | 84.36 |
| | Pooled Flair PubMed | 32 | 84.92 | 89.9 | 87.34 |
| | Transformer Bert | 16 | 76.28 | 82.74 | 79.37 |
| | Transformer BioBert | 32 | 82.58 | 89.58 | 85.94 |
| MedLine | ELMo Original | 8 | 58.8 | 54.59 | 56.61 |
| | ELMo PubMed | 16 | 67.29 | 72.26 | 69.69 |
| | Pooled Flair Mix | 16 | 52.78 | 51.01 | 51.88 |
| | Pooled Flair PubMed | 8 | 64.81 | 62.64 | 63.71 |
| | Transformer Bert | 16 | 48.44 | 55.48 | 51.72 |
| | Transformer BioBert | 16 | 60 | 68.46 | 63.95 |

With respect to the results tested with the DrugBank corpus using classic word embeddings, the best findings for each model were achieved using the general word embedding CC (see Table 4). In terms of F1-scores, the CRF + CC model achieved a percentage that was a little higher than the CRF + Pyy model. This result is similar to one presented by *Segura-Bedmar, Suárez-Paniagua & Martnez (2015)*, where in order to improve the performance of the BioNER task, they achieved 1 pp higher when using general instead of specific word embeddings. However, they did not present an extensive comparison between embeddings and also, they appended several additional features to the word embeddings which make difficult to observe the real behavior of the embeddings. As Table 4 shows, this behavior was also maintained in the BiLSTM (BiLSTM + CC: 86.45% vs. BiLSTM + Pyy: 83.68%), and the BiLSTM-CRF models as well (BiLSTM-CRF + CC: 88.32% vs. BiLSTM-CRF + Pyy: 84.79%). According to *Unanue, Borzeshi & Piccardi (2017)*, the performance difference between these word embeddings could be attributable to their respective word coverages. With regards to biological domains, it is to be expected that a specific word embedding will have better word coverage than a general form and, therefore, better performance. But this behavior was not reflected in the case of DrugBank (see Table 5), where the specific word embedding had better coverage, but not the best performance. This experimental finding shows, on the one hand, that word coverage is not a determining factor in choosing a form of word embedding and, on the other, that specific word embedding is not always better than the general form, when using classic word embeddings. Therefore, general word embedding should not be discarded as a possible feature for biological named entity recognition.

In contrast to our observations of DrugBank, when using classic word embeddings in MedLine (see Table 4) the specific word embedding Pyy achieved the best scores for the BiLSTM and BiLSTM-CRF models (BiLSTM + CC: 60.98% vs. BiLSTM + Pyy: 63.78%; BiLSTM-CRF + CC: 65.37% vs. BiLSTM-CRF + Pyy: 71.55%). This is consistent with the

findings in *Batbaatar & Ryu (2019)*, where all the specific classic word embeddings, in combination with other characteristics, obtained the best scores for biological domains.

In addition, we performed a comparison of the results for every model×classic-embedding combination between the corpora. This analysis revealed that DrugBank had the best scores; for instance, the combination of BiLSTM-CRF with CC embedding over DrugBank obtained an F1-score of 88.32%, while using MedLine and the same model×classic-embedding the F1-score obtained fell to 65.37%. In the case of the combination of BiLSTM-CRF with Pyy embedding over DrugBank, the F1-score was 84.79%, while using MedLine and the same model × classic-embedding achieved an F1-score of 71.55%. This highlights a difference of 22.95 pp and 13.24 pp, respectively, even though both corpora were used to recognize drug names. While these differences could be a consequence of the size of the training set, as hypothesized by *Unanue, Borzeshi & Piccardi (2017)*, Fig. 2 shows that these results represent only part of the differences in the scores between the corpora. Thus, the remaining difference could be attributed to at least two other elements; one involving the source and number of tokens used to train the word embeddings, the other regarding the origin and content of the corpora. While 840 billion general web tokens were used to train CC, only 5.5 billion specialized PubMed and PubMed Central tokens were used for Pyy. The resulting sizes of the vocabularies were 2.2 million for CC and 4 million for Pyy, showing that the general CC has a shorter, less specialized vocabulary generated with more extensive training. For this reason, while CC coverage may be shorter, thanks to its extended training it could perform better, at least with DrugBank. The MedLine corpus, meanwhile, comes from PubMed abstracts and the Pyy word embedding from PubMed and PubMed Central. This led to the hypothesis of a relation between the MedLine corpus and Pyy embedding that is not seen in DrugBank because it comes from documents in the DrugBank database (*Herrero-Zazo et al., 2013*).

With the goal of reaching a deeper understanding of the characteristics of these classic word embeddings, the results of the third experiment (see Table 6) show that the semantic similarities captured by the specialized word embedding Pyy are more akin to the judgments of the human experts than those obtained by the general word embedding CC. Despite the fact that Pyy embedding has better cohesion for biomedical entities, CC embedding is better for DrugBank, so it is necessary to consider that there may be a relationship between the corpus selected and the form of word embedding applied.

Unlike classic word embeddings, where general embeddings can perform better than specific ones, in the contextualized word embeddings the best performances for both corpora (DrugBank and MedLine) are achieved using specific embeddings. As Table 7 shows, the differences between specific and general embeddings in the Medline corpus (ELMo: 13.08 pp, Pooled Flair: 11.83 pp, Transformer: 12.23 pp) are greater than those observed in the DrugBank corpus (ELMo: 4.02 pp, Pooled Flair: 2.98 pp, Transformer: 6.57 pp). As mentioned above, MedLine corpus was obtained from PubMed, so this can provide an advantage to specific contextualized embeddings since they have been trained in this type of biomedical text and they can modify the vector of each word dynamically. However, this property of modifying the vectors dynamically seems counterproductive for general contextualized embeddings since they were training over colloquial texts.

On the other hand, comparing the performance of the three contextualized word embeddings on the BioNER task, it can be observed that the ELMo PubMed obtain the best scores for both corpus, this is, 88.28% for DrugBank and 69.69% for MedLine. However, to generalize this statement extensive experimentation on more corpora is required.

Regarding a cross-comparison between classic and contextualized embeddings, the results of the first experiment might not be directly comparable with the results of the fourth experiment because they differ in the implementation of the BiLSTM-CRF network, although the parameters have been set as similar as possible. Despite this fact, the same conclusions can be reached from the first experiment using the Flair library. Likewise, the present study has only compared contextualized embeddings with extrinsic evaluation on the BioNER task. An intrinsic evaluation of contextualized embeddings like the one carried out with classic embeddings is not possible since each word can have multiple vectors according to its context.

In summary, regarding to classic word embeddings, although the specific word embeddings showed the best biomedical word coverage and best semantic relationship among medical entities, they are not always a better option than general word embeddings. The latter, therefore, could be considered as a possible automatic feature extractor for the BioNER task based on the understanding that general embeddings could be used when it is necessary to extend the model to classify non-biomedical entities as well. However, in regard to contextualized word embeddings, the best option to use is a specialized one.

## CONCLUSIONS

This article presents the results of an evaluation of different word embeddings trained from general and specific sources. The assessment was performed in four ways; first, by focusing on the BioNER task and testing the CC and Pyy classic word embeddings (general and specific, respectively) on the DrugBank and MedLine corpora using the CRF, BiLSTM, and BiLSTM-CRF algorithms. In the second approach, we concentrated on the corpus size by applying multiple reductions to the corpora. The third experiment used an intrinsic evaluation of the classic word embeddings focused on the semantic similarity between pairs of words from four different datasets: Pedersen, Hliaoutakis, MayoSRS, and UMNSRS. In the last approach, we compared the performance of general vs. specific contextualized word embeddings using the BiLSTM-CRF algorithm with three state-of-the-art embeddings: ELMo, PooledFlair, and Transformer. The following conclusions can be drawn from the results of the present evaluation. First, the model with the best performance for drug-named recognition using only word embeddings as features was BiLSTM-CRF, while the model with the lowest performance was CRF. Second, using classic word embeddings, the best scores were not always obtained with specific embeddings, since better performance was observed using the general word embedding CC in the DrugBank corpus for all three algorithms. This result was apparent even though general word embeddings have less word coverage than specific ones. Third, the quality of performance decreases as the training corpus is reduced, as was clear for the DrugBank corpus. However, even in that case, the performance achieved with the general CC embedding was better than the result obtained with the specific Pyy embedding. Fourth, classic general word embeddings can be considered

characteristic for BioNER, even though their semantic similarity between medical words is of a lower order than that of a classic specific embedding. Fifth, for the selection of contextualized word embeddings on the BioNER task, it would be better to choose the specific ones. Finally, among the contextualized embeddings and under the same conditions, the ELMo embeddings achieved the best results for DrugBank and MedLine, indicating that these embeddings are a viable feature for BioNER.

As a future direction for research, our plan is to evaluate the re-trained word embeddings; that is, those generated by joining general and specific word embeddings. This approach will require an extensive selection of word embeddings, testing different methods for evaluating the difference between them (such as clustering). In regard to contextualized word embeddings, it would be important to elucidate a method to evaluate them intrinsically to determine if this novel word representations are more correlated to the gold standard. Finally, an experiment will be designed to demonstrate the possible performance relationship between a particular form of word embedding and the corpus used to train the models for BioNER.

### Funding
CONACyT provided a scholarship grant (CVU 999749) to Rigo E. Ramos-Vargas. The funders had no role in study design, data collection and analysis, decision to publish, or preparation of the manuscript.

### Grant Disclosures
The following grant information was disclosed by the authors:
CONACyT: CVU 999749.

### Competing Interests
The authors declare that they have no competing interests.

### Author Contributions
- Rigo E. Ramos-Vargas conceived and designed the experiments, performed the experiments, analyzed the data, performed the computation work, prepared figures and/or tables, and approved the final draft.
- Israel Román-Godínez conceived and designed the experiments, analyzed the data, prepared figures and/or tables, authored or reviewed drafts of the paper, and approved the final draft.
- Sulema Torres-Ramos conceived and designed the experiments, analyzed the data, prepared figures and/or tables, authored or reviewed drafts of the paper, and approved the final draft.

### Data Availability
Code and data are available in the Supplemental Files.

## Supplemental Information

Supplemental information for this article can be found online at http://dx.doi.org/10.7717/peerj-cs.384#supplemental-information.

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
