# Peer review of "Comparing general and specialized word embeddings for biomedical named entity recognition"

_PeerJ Computer Science, doi:10.7717/peerj-cs.384_

## Round 0.1 · original submission · Major Revisions

As mentioned by all of the reviewers, the paper is well structured and well written. There were, however, concerns about the novelty of the contribution (Reviewer 2) and the comprehensiveness of the evaluation (Reviewer 3).

I recommend that the paper be reconsidered for acceptance after a slightly expanded evaluation that used at least one more modern, contextual word embedding techniques. Reviewer 1 had some concrete suggestions along these lines.

Reviewer 1 ·

Basic reporting

This paper presents a performance evaluation of different word embeddings for the task of named-entity recognition (NER) in the biomedical domain.

In brief, the paper compares two word embeddings (GloVe word embeddings trained on a Common Crawl corpus and word2vec word embeddings trained on PubMed abstracts and PubMed Central articles) over three experiments.

The paper is clear, well organised, and well presented throughout.

However, it is affected by a major shortcoming which is the very limited scope of its evaluation. As it stands, I do not believe this paper can be recommended for acceptance, but the evaluation could be expanded to become appropriate.

Experimental design

The experimental design and reporting are appropriate. The experiments are organised to evaluate the impact of 1) the type of word embeddings and 2) the training set size on NER accuracy, and the impact 3) of the type of word embeddings on the correlation between the embeddings’ pairwise similarity and human-annotated pairwise similarity. Three NER models are considered.

The only remark is that when the authors mention differences between scores, they should use "percentage points" (pp) as unit, not "%" (e.g., "1.68% and 1.35%" should be: "1.68 pp and 1.35 pp". The differences are not percentages; the absolute values are.

Another minor remark is about the name used for the Common Crawl word embeddings: there exist other popular Common Crawl word embeddings, obtained with different algorithms; for instance, fastText's. The authors should say upfront that these are GloVe Common Crawl word embeddings. The Pyysalo description is ok as is.

Validity of the findings

The current findings seem valid. However, as I have already mentioned in the opening field, the scope of the evaluation is too limited to make it contemporary and comprehensive. The manuscript is organised in the vein of its main reference, Unanue at al. [2017], but three years have passed since, and many other word embeddings and models have been proposed. To gain an idea, one can easily browse sites such as NLP-progress (http://nlpprogress.com/english/named_entity_recognition.html) and Papers with Code (https://paperswithcode.com/task/named-entity-recognition-ner).

An evaluation of appropriate scope is, for instance, the manuscript’s own reference Batbaatar and Ryu (2019) which has compared 14 word embeddings.

To make the evaluation sizable and interesting, yet not overwhelming for the authors, I recommend the following extensions:

1) LUKE. The official code can be found at the link on NLP-progress. The authors should use the embeddings produced by LUKE with their models, and possibly also with the provided NER classifier. The model is pretrained over Wikipedia pages. Of course, fine tuning over Common Crawl and PubMed would further value to the comparison.

2) Flair https://github.com/flairNLP/flair. Its Python library is straightforward to use; please see the example.

My recommendation is not prescriptive in terms of specific methods. I simply recommend expanding the evaluation to embrace more modern, contextualised word embeddings to make it contemporary.

Additional comments

I do not have any additional comments.

Reviewer 2 ·

Basic reporting

The paper presents an evaluation of how pre-trained vectors of general corpora can be helpful in BioNER. Despite that, many previous works already proved that domain-specific trained embeddings are better than general ones. The paper is well written, and the structure is connected and presented in the right way. The authors were able to present a good list of related works. Figures and Tables were all clear.

Experimental design

no comment

Validity of the findings

no comment

Additional comments

- Grammar errors:
There was only one grammar error in line 277, where the word "Table" was repeated two times. Just remove the repeated word.
- Results and paper contribution:
Results were confused a little bit. The authors reported that some previous works already made this research contribution, such as Segura-Bedmar et al. (2015) “who found that the performance achieved with the DrugBank corpus and the CRF model was 1% higher when using general instead of specific word embedding.” So, in terms of general vs domain-specific embeddings, Segura already proved that general is sometimes a good source for NER systems. The only contribution in this paper is that the BiLSTM-CRF was better than the CRF. This contribution can fit to a short paper or a poster in a conference more than a journal.

·

Basic reporting

The article is written in English that is easy to understand. The overarching goal of the project is clear. However, since not everyone is familiar with the term "word embedding" or "biomedical named entity recognition", concise explanations of the terms would be helpful. Also, it's critical to make it clear to a general reader how the training sets and validation sets are obtained, and how the class labels are defined and assigned for a sample or data point.

Experimental design

As stated in the "basic reporting" section above, the actual experimental design is probably OK. However, explanations need to be given to the two basic terms, the obtainment of training sets and the validation sets, and the class labels.

While it's OK to use precision/recall for performance measurement, it's not clear why it is preferred over sensitivity/specificity/AUC. Is it because true negatives (TNs) are not easy or impossible to label? If so, it would help to explain it briefly.

A minor issue is about the description about precision, recall, and F1-score between line 223 and 224. These terms are well known. I am not sure it's worth putting them in the manuscript. Instead, references to these terms can be given in the article.

Validity of the findings

If the training sets and validation sets are constructed correctly, and the class labels are assigned in the right way, the F1-score, the performance plot, and the correlation between the machine results and the human results are convincing enough.

Additional comments

This is a reasonable research project with potentially publishable results. More explanations to the terms, training sets, validation sets, and class labels are critical for readers to assess the scientific basis of the research.

---

## Round 0.2 · accepted · Accept

The authors have addressed all review comments so the paper is suitable for publication.